# Disruption of the Chitin Biosynthetic Pathway Results in Significant Changes in the Cell Growth Phenotypes and Biosynthesis of Secondary Metabolites of *Monascus purpureus*

**DOI:** 10.3390/jof8090910

**Published:** 2022-08-27

**Authors:** Meng Shu, Pengxin Lu, Shuai Liu, Song Zhang, Zihan Gong, Xinru Cai, Bo Zhou, Qinlu Lin, Jun Liu

**Affiliations:** Hunan Key Laboratory of Grain-Oil Deep Process and Quality Control, National Engineering Research Center of Rice and Byproduct Deep Processing, College of Food Science and Engineering, Central South University of Forestry and Technology, Changsha 410004, China

**Keywords:** *Monascus purpureus*, chitin synthase, morphology, stress tolerance, secondary metabolite

## Abstract

In this study, the gene *monascus*-5162 from *Monascus purpureus* LQ-6, identified as chitin synthase gene VI (*chs*6), was knocked out to disrupt the chitin biosynthetic pathway and regulate the biosynthesis of *Monascus* pigments (MPs) and citrinin. The results showed that the aerial hyphae on a solid medium were short and sparse after the deletion of *chs*6 in *M. purpureus* LQ-6, significantly reducing the germination percentage of active spores to approximately 22%, but the colony diameter was almost unaffected. Additionally, the deletion of *chs*6 changed the mycelial morphology of *M. purpureus* LQ-6 during submerged fermentation and increased its sensitivity to environmental factors. MP and citrinin biosynthesis was dramatically inhibited in the recombinant strain. Furthermore, comparative transcriptome analysis revealed that the pathways related to spore development and growth, including the MAPK signaling pathway, chitin biosynthetic pathway, and regulatory factors *Lae*A and *Wet*A genes, were significantly downregulated in the early phase of fermentation. The mRNA expression levels of genes in the cluster of secondary metabolites were significantly downregulated, especially those related to citrinin biosynthesis. This is the first detailed study to reveal that *chs*6 plays a vital role in regulating the cell growth and secondary metabolism of the *Monascus* genus.

## 1. Introduction

The filamentous fungus *Monascus* is an important fermentative microorganism that has been utilizing external substrates to synthesize several secondary metabolites, including *Monascus* pigments (MPs), monacolin K, and citrinin, in Asia for centuries. Generally, research on the regulation of the production of these secondary metabolites has been mainly carried out using biochemical technology [1,2,3], breeding technology [4,5,6], response surface optimization [7,8], and conventional genetic engineering [9,10,11]. It has been reported that the biosynthesis of pigment and monacolin K may be closely related to the mycelial morphology of the *Monascus* genus during the submerged fermentation. Chen et al. reported that altered cellular morphology stimulated the biosynthesis of *Monascus* pigment and secretion during extractive fermentation [12]. In addition, researchers have investigated the correlation between pigment production and mycelial morphology in the extractive fermentation of *Monascus* anka GIM 3.592 [13]. In addition, Lv et al. controlled the fermentation conditions to change the mycelial morphology of *Monascus purpureus*, thereby enhancing the production of natural yellow pigments during liquid-state fermentation [14]. Furthermore, to efficiently produce yellow pigments, the hyphal morphology and cell activity of *M. purpureus* were influenced by the addition of surfactants [15].

It is well known that the mycelia morphology of filamentous fungus can be divided into three categories, i.e. dispersed hyphae, mycelia clumps, and mycelia pellets, in submerged fermentation [16]. Changes in mycelial morphology influence the rheological properties of the fermentation broth and the transfer of heat and mass, which determine the production and productivity of the target products [17]. Recently, morphological regulatory genes have been reported to have a significant influence on cell growth and secondary metabolites in *Monascus*. For instance, the overexpression of the global regulator *Lae*A alters the mycelial morphology of *M. purpureus* and significantly increases monacolin K production in submerged fermentation [18]. Similarly, it has been reported that the *Lae*A gene regulates not only the biosynthesis of pigments and citrinin but also the mode of cell differentiation [19]. It has also been reported that the transcription factors *Brl*A and *Wet*A have an insignificant effect on the morphology, size, number, structure, and germination of conidia but could significantly regulate sexual development and the production of secondary metabolites of *Monascus* [20]. These results suggest that the morphological regulation genes play a vital role in regulating cell growth, phenotypes, and biosynthesis of secondary metabolites in *Monascus*.

The rise in metabolic engineering of the morphology of filamentous fungi in submerged fermentation illustrate the relationship between cell growth, phenotypes, and the target product. Hundreds of regulation genes have been unearthed as the targets to perform the metabolic engineering of the morphology for regulating the biosynthesis of target metabolites. Chitin, a major structural component of the fungal cell wall, is synthesized by chitin synthase (*chs*) and has become a high-profile target for investigating the morphology and metabolism of filamentous fungi. For example, Liu et al. silenced the class III *chs* gene to change the morphology of *Penicillium chrysogenum*, resulting in the overproduction of penicillin [21]. The relationship between *Aspergillus oryzae* morphology and α-amylase production was analyzed by deleting the *chs*B gene [22]. In addition, it has been reported that enzymes differentially affect fungal growth, stress tolerance, cell wall integrity, and virulence [23]. However, the morphological regulation genes and the regulation of the biosynthesis of secondary metabolites by metabolic engineering of the morphology of *Monascus* still lack an overall and rational understanding.

Therefore, in this study, the members of chitin synthases were annotated based on genome-wide sequencing, and the *chs*6 gene was selected as a target to study the regulatory effect of *chs*6 on cell growth and biosynthesis of secondary metabolites in *M. purpureus*. Furthermore, we have performed transcriptome analysis to elucidate the regulatory mechanism of chitin synthases in *M. purpureus*.

## 2. Materials and Methods

### 2.1. Chemicals and Enzymes

Restriction and other enzymes were purchased from Takara Bio (Shiga, Japan). The one-step cloning kit for ligation and the gel extraction kit were purchased from Vazyme Biotech Inc. (Nanjing, China), and the plasmid mini kit was purchased from Omega Bio-Tek (Norcross, GA, USA). All other chemicals were commercially available.

### 2.2. Strains and Cultivation

All the strains and plasmids used in this study are listed in Appendix A. The strains *M. purpureus* LQ-6 (the parent strain) and *M. purpureus* Δ5162 and cp-5162 (recombination strains) were cultured on potato dextrose agar (PDA) medium with 100 μg/mL hygromycin B or 50 μg/mL G418 at 30 °C in the dark. Luria-Bertani medium with 50 μg/mL ampicillin was used for culturing *Escherichia coli* DH5α at 37 °C and 180 rpm.

### 2.3. Bioinformatics Analysis of Gene of Monascus_5162

The nucleotide and amino acid sequences of *monascus*_5162 were submitted to the screening public database using National Center for Biotechnology Information (NCBI) BLASTp (https://blast.ncbi.nlm.nih.gov/Blast.cgi, accessed on 1 July 2022) for homology analysis. Domain prediction was performed with NCBI-CDD (NCBI Conserved Domain Search (nih.gov, accessed on 1 July 2022)).

### 2.4. Construction of Gene Deletion and Complementation Plasmids

Primers used in this study are listed in Appendix A. To construct the deletion plasmid pXS-5162, the 1.5 kb upstream and downstream homology arms of the gene *monascus*_5162 were amplified by PCR from the genomic DNA of LQ-6 using the primers 5162-up-F/R and 5162-dn-F/R with 20 bp homologous regions to the pXS-G418 plasmid. To construct the complementation plasmid pBA-5162, a 5.5 kb *monascus*_5162 open reading frame (ORF) region was amplified by PCR from the genomic DNA of LQ-6 using primers pBA-5162-F/R and ligated to the pBARGPE1-Hygro plasmid. These construction processes are shown in Appendix A. The deletion and complementation vectors were constructed to obtain recombinant plasmids according to our previously reported method [24].

### 2.5. Preparation and Transformation of Protoplast

The methods of preparation and transformation of the protoplast from the parent strain were reported in our previous study [3]. *M. purpureus* LQ-6 was cultured in PDB medium at 30 °C. After approximately 2 days of static culture, the germinated hyphae were collected in a sterile 2 mL centrifuge tube through centrifugation at 8228× *g* for 10 min, washed twice with 2 mL osmotic medium, transferred into 1.5 mL osmotic medium containing lysozyme, and incubated at 37 °C for 3 h. The protoplasts were then obtained by filtration using an injection syringe containing cotton. Next, trapping and STC buffers used for washing were added, and then the protoplasts were centrifuged at 4629 × *g* and 4 °C for 5 min. Protoplasts were diluted with fresh STC buffer to an appropriate concentration of 1 × 10^7^ protoplasts/mL. Then, 200 μL of the protoplast solution and 10 μg linearized pXS-5162 were mixed gently and incubated on ice for 60 min, and 1.25 mL PEG 6000 solution and 5 mL PDB-sorbitol were added successively without resistance and incubated at 30 °C for 15 h. The final mixture was coated on PDA with 50 μg/mL of G418 and cultured in the dark at 30 °C for 5 days.

### 2.6. Complementation of Gene Monascus_5162

The vector pBA-G418 digested with *EcoRI* and *XmaI* was ligated with the *monascus*_5162 fragment using a one-step cloning kit, and the pBA-5162 plasmid was transformed into the gene deletion mutant strain by PEG transformation. The final mixture was coated on PDA plates with 100 μg/mL of hygromycin B.

### 2.7. Confirmation of the Mutation Strains

After consecutive and stable passages, colony PCR was used to confirm the positive transformations and acquire the gene deletion mutant strain (*M. purpureus* Δ5162) and gene complementation strain (*M. purpureus* cp-5162) using the primers 5162-T-F/R and 5162:cp-T-F/R, respectively.

### 2.8. Image Analysis

The colony morphologies of *M. purpureus* LQ-6 and Δ5162 on PDA plates were monitored using a digital camera (Canon EOS 80 D, Tokyo, Japan), and the colony diameters were measured daily. Mycelial pellets were monitored using a digital camera (Canon EOS 80 D), and free hyphae were monitored using a biological microscope (Olympus, Tokyo, Japan). Samples of *M. purpureus* LQ-6 and Δ5162 were treated according to a previously described method (Liu, Chai et al., 2019), and images were observed using a transmission electron microscope (TEM) (Hitachi H-7650, Hitachi, Japan).

### 2.9. Stress Tolerance Analysis

A spore suspension (10^6^–10^7^ spores/mL) was obtained by washing the PDA plate containing *M. purpureus* strains with sterile water. Then, 3 μL of the spore suspension was placed on PDA plates containing ethanol (0%, 2%, 4%, 8%, 10%), lactic acid (0%, 2%, 4%, 8%, 10%), sodium chloride (0 M, 0.3 M, 0.6 M, 0.9 M, 1.2 M), and hydrogen peroxide (0 mM, 2 mM, 3 mM, 4 mM, 8 mM) to observe the changes in colony morphology.

### 2.10. Submerged Fermentation

For submerged batch-fermentation, 10 mL of the spore suspension (10^6^–10^7^ spores/mL) was inoculated into 90 mL fermentation medium in 250 mL conical flasks containing 60 g/L glucose, 20 g/L peptone, 1 g/L yeast extract, 0.5 g/L MgSO_4_·7H_2_O, 0.05 g/L FeSO_4_·7H_2_O, and 0.1 g/L ZnSO_4_·7H_2_O at neutral pH and incubated for 10 days at 150 rpm at 30 °C. For kinetic analysis, the volume of the fermentation broth was expanded to 300 mL in 1000 mL conical flasks for submerged fermentation.

Submerged fermentations with different concentrations (20 mg/L, 50 mg/L, 100 mg/L, 200 mg/L, and 300 mg/L) of polyoxin B were performed as described above for submerged batch-fermentation in 250 mL Erlenmeyer flasks.

### 2.11. Determination of Metabolites

The processing and measurement of citrinin and MPs content, residual glucose concentration, biomass, and electrical conductivity were conducted according to our previous report [24]. The collected fermentation broth was first scanned using a microplate at wavelengths of 370 and 410 nm for yellow MPs (YMPs) and 510 nm for red MPs (RMPs), corresponding to the maximum absorption peaks.

The chitin content of *M. purpureus* strains was assayed according to a previously reported method [25]. The extracellular alkine phosphatase (AKP) activity of *M. purpureus* strains was determined using a UV/Vis spectrophotometer (UV-752 N) with the AKP kit (Solarbio Science and Technology, Beijing, China) following the manufacturer’s instructions. Enzyme activity is expressed as U/g protein. Each experiment was repeated three times. Numerical data are presented as mean ± standard deviation.

### 2.12. Transcriptome Analysis

Fresh mycelia of parent strain *M. purpureus* LQ-6 and mutant strain Δ5162 (three replicates per sample, a total of 12 samples, named CK-4d-1, CK-4d-2, CK-4d-3, CK-7d-1, CK-7d-2, CK-7d-3,and MT-4d-1, MT-4d-2, MT-4d-3, MT-7d-1, MT-7d-2, MT-7d-3, respectively) were collected at 4 d and 7 d and sent immediately to Gene Denovo Biotechnology Co. (Guangzhou, China) for mRNA sequencing. Total RNA was extracted using the Trizol reagent kit (Invitrogen, Carlsbad, CA, USA) according to the manufacturer’s protocol. RNA quality was assessed on an Agilent 2100 Bioanalyzer (Agilent Technologies, Palo Alto, CA, USA) and checked using RNase free agarose gel electrophoresis. After enrichment by Oligo(dT) beads and synthesized and purified second-strand cDNA fragments, they were purified with QiaQuick PCR extraction kit(Qiagen, Venlo, The Netherlands), end-repaired, combined with poly(A), and ligated to Illumina sequencing adapters. Finally, the ligation products were size-selected by agarose gel electrophoresis, PCR amplified, and sequenced using Illumina HiSeq6000.

After filtering of Clean Reads by fastp (version 0.18.0), RNAs’ differential expression analysis was performed by DESeq2 software between two different groups. The genes/transcripts with the parameter of false discovery rate (FDR) below 0.05 and absolute fold change ≥ 2 were considered differentially expressed genes (DEGs). DEGs were subjected to Gene Ontology (GO) function and Kyoto Encyclopedia of Genes and Genomes (KEGG) pathway enrichment analyses.

### 2.13. Accession Numbers

All raw RNA-seq data were deposited in the NCBI database using the BioProject accession number: PRJNA785330. The gene *monascus_5162* is located in the genome sequence of *M. purpureus* LQ-6 with the accession number PRJNA503091.

### 2.14. Statistical Analysis

Each experiment was performed at least in triplicate, and the results are shown as the mean ± standard deviation (SD). Statistical analyses were performed using the SPSS Statistics 23 (SPSS, Chicago, IL, USA). Data were analyzed by one-way ANOVA, and tests of significant differences were determined using Tukey multiple comparison or Student’s *t*-test at *p* < 0.05.

## 3. Result and Discussion

### 3.1. Functional Annotation of Gene Monascus_5162

According to the data of whole-genome sequencing of *M. purpureus* LQ-6 (deposited into the NCBI database with an accession number of PRJNA503091), eight genes, *monascus*_2400, *monascus*_2508, *monascus*_2563, *monascus*_2765, *monascus*_2870, *monascus*_4382, *monascus*_5161, and *monascus*_5162, presumptively encode chitin synthase. We used the BLAST online tool in the NCBI to confirm the annotation of *monascus*_5162 (5480 bp, 1755 aa), and the results showed that the nucleotide and amino acid sequences of *monascus*_5162 were homologous to those of the chitin synthase protein at locus no. ACV03807, 1755 aa, and the chitin synthase protein at locus no. FJ643483, 5268 bp, from *M. ruber* M7 with 99.97% and 99.89% identity and a query cover of 96% and 100%, respectively. In addition, it has previously been reported that members of chitin synthase fall into seven discernable classes (class I–VII), generally containing six to ten genes in filamentous fungi, and their functional significance is indistinct [23,26]. The classes I–III share a common catalytic domain surrounded by a hydrophilic N-terminal region and a hydrophobic C-terminal region, classes IV–VII conventionally contain a cytochrome b5-binding domain, and classes V and VI both consist of a C-terminal chitin synthase domain and an N-terminal myosin-motor domain [27]. It has been reported that the myosin-motor domain of class VI chitin synthases does not contain the consensus ATP-binding motifs of p-Loop, Switch I and Switch II, which are thought to be essential for ATPase and motor activities but are conserved in the myosin-motor domain of class V chitin synthases [28]. Conserved domain analysis using the batch web CD-search tool in NCBI showed that the gene *monascus*_5162 contains a myosin-motor domain (cd14879, 22–587 aa), cytochrome b5-binding domain (cl02041, 793–850 aa), and chitin_synth_2 domain (pfam03142, 1037–1539 aa). Furthermore, the protein chitin synthase from *A**. nidulans* FGSC A89, *Csm*B (BAE78841), is annotated as *chs*6 (https://www.ncbi.nlm.nih.gov/protein/86355213; accessed on 15 February 2006) [28]. Then, the tertiary structures of protein *monascus*_5162 and *Csm*B were predicted and comparatively analyzed by using the phyre2 web server, and the result showed that the spatial structures of their proteins were almost the same, except for the longer C-terminal tail in *Csm*B and 2 hits in *monascus*_5162 (Figure 1). Thus, gene *monascus*_5162 was identified as *chs*6 in *M. purpureus* LQ-6.

### 3.2. Regulation Effect of Chs6 on Cell Growth Phenotypes

It has been reported that *chs*6, *chs*4, and *chs*7 are essential for the normal growth and pathogenicity of *Botrytis cinerea* [27,29]. In addition, the sensitivity of the *chs*6-deleted mutant strain to cell-wall-disturbing chemicals, including calcofluor white, 5-bromo-4-chloro-3-indolylphosphate, sodium dodecyl sulfate, and NaCl, was changed [27]. As described in the introduction, members of chitin synthase play a vital role in regulating the cell development and the mycelial morphology of filamentous fungi. In this study, we investigated the effect of *chs6* on sensitivity and tolerance to environmental factors, development, cell growth, mycelial morphology, and cell wall structure.

We found that the *Monascus* genus is extremely sensitive to lactic acid, and the cell growth of *M. purpureus* LQ-6 was distinctly inhibited on PDA plates with the addition of 2% (*v/v* lactic acid. However, there was a strong tolerance (growth still appeared at 10% lactic acid, Figure 2A). In addition, the germination time of the LQ-6 strain cultivated on PDA medium with 8% lactic acid was prolonged from the 2nd day to the 4th day after knocking out *chs*6, and the minimum inhibition concentration (MIC) of lactic acid against *M. purpureus* LQ-6 and Δ5162 was >10% and 8%, respectively. Figure 2B shows that the parent strain has a strong tolerance to ethanol, but the germination time of *M. purpureus* Δ5162 was prolonged from the 3rd day to the 7th day in the presence of 8% *(v/v*) ethanol, and the cell growth was completely inhibited at a concentration of 10%; under the same conditions, the inhibition rate of colony diameter of the parent strain was approximately 37.39% at 8 days. The sensitivity to H_2_O_2_ was significantly increased in the recombinant strain compared with that in the parent strain, and no visible growth of *M. purpureus* Δ5162 was apparent in response to 3 mM H_2_O_2_, but the colony diameter of the LQ-6 strain in response to 8 mM H_2_O_2_ weakly decreased by 23.48% at 8 days (Figure 2C). However, the regulation effect of *chs*6 in the sensitivity to NaCl (the concentration range of 0–1.2 mol/L) was insignificant, and the MICs of NaCl were both at 1.2 mol/L against *M. purpureus* LQ-6 and Δ5162 (Figure 2D). The phenomenon (the cell growth phenotypes on the PDA plates are shown in Appendix A) demonstrated that the sensitivities to ethanol, lactic acid, and H_2_O_2_, but not to NaCl, consistently increased after the deletion of *chs*6 in *M. purpureus* LQ-6.

Furthermore, we investigated the changes in the biomass and morphology after deletion of *chs*6 in *M. purpureus* LQ-6 during submerged fermentation. Although there were almost no significant changes in the morphology of mycelium (except that the aerial hyphae on PDA medium were short and sparse) and the growth rate of colony diameter (conidial germination rate) between the *chs*6-deleted and LQ-6 strains (Figure 3(A1–A3)), the mycelial morphology and cell wall structure were significantly regulated by *chs*6 during submerged fermentation. It has been reported that the conidial germination rate of *Metarhizium acridum* was significantly decreased (slower germination) after knocking out gene classes I, III, IV, VII, compared with that of the wild type, but was weakly reduced following the knock-out of class IV genes alone (which was the same as our results) [23]. Selected photographs suggest that mycelium pellet formation differed between the two strains. The surfaces of the LQ-6 pellets were smooth and small on the 3rd day, but *M. purpureus* Δ5162 was in the form of free mycelium. On the 6th day, the pellets of *M. purpureus* Δ5162 were rougher and much fleecy than those of the parental strain (Figure 3(B1,B2)). Similarly, our previous study found that the surfaces of the pellets exhibited a slightly hairy appearance and lower proportions of dispersed *Monascus* hyphae after the deletion of *erg*4A and *erg*4B, which encode C-24(28) sterol reductase involved in catalyzing the biosynthesis of ergosterol [24]. This illustrated that the disruption of the cell wall and membrane structures would change the mycelial morphology in submerged fermentation. Furthermore, the biomass of *M. purpureus* LQ-6 and Δ5162 reached the maximum values of 21.45 g/L on the 9th day and 17.24 g/L on the 10th day, respectively (Figure 3(B3)). The maximum biomass decreased by 19.63% after disrupting the chitin synthetic pathway by deleting *chs*6 in submerged fermentation, but the growth rate of the colony diameter was unchanged. Furthermore, the germination rates of the conidia of the two strains on the PDA medium were analyzed (shown in Appendix A). We found that the conidial germination ratio of the LQ-6 strain was approximately 89% at 2 days and 91% at 5 days, but that of the mutant strain was approximately 22% at both 2 and 5 days. These results illustrate that the reduced maximum biomass in submerged fermentation most likely imputed the impaired germination ratio of conidia by deletion of the *chs*6 gene in *M. purpureus* LQ-6.

Figure 3C shows that some balloon-tip-like structures in the hyphae of *the chs*6-deleted mutant strain appeared on day 3, and the hyphae were longer and had low branch efficiency. It has been reported that the hyphal elongation rates were significantly increased after the knockout of class I, II, IV, and VI chitin synthase genes in *M. acridum* [23]. These results indicate that the *chs*6 gene plays a significant role in the tip extension of filamentous fungi. In addition, the number of tip balloons further increased, but the number of cleistothecia was significantly reduced at day 6 in *M. purpureus* Δ5162 compared with that in the parent strain. This illustrated that class VI chitin synthase could stimulate sexual development in *Monascus* and regulates the formation of mycelial pellets. Furthermore, the results of TEM revealed that the cell wall thickness showed significant changes, and the cell wall was thinner in the mutant strain than in the wild type. In addition, the organelles in the wild-type strain were still intact at 6 days but were distorted or broken after the deletion of the *chs*6 gene (Figure 3D). In conclusion, our work not only clarified that the *chs*6 gene has a significant regulatory effect on *Monascus* growth phenotypes but also proved that it is required for normal growth.

### 3.3. Regulation Effect of Chs6 on Monascus Pigment and Citrinin Biosynthesis

MPs are a mixture of azaphilones and are mainly composed of YMPs, orange MPs, and RMPs, which generally have maximum absorption peaks at 330–450 nm, 460–480 nm, and 490–530 nm, respectively [30]. In the present study, we found that three absorption peaks at 370, 410, and 510 nm were obtained from the scan of the fermentation broth from the recombination strains (*M. purpureus* Δ5162 and *M. purpureus* cp-5162) and the parent strain LQ-6, for measuring the concentrations of YMPs (at 370 nm and 410 nm) and RMPs (at 510 nm). From Figure 4, after the deletion of the *chs*6 gene in *M. purpureus* LQ-6, the total production of YMPs_-370 nm_, YMPs_-410 nm_, and RMPs decreased to 15 U/mL, 16 U/mL, and 9.24 U/mL from 51.17 U/mL, 52.40 U/mL, and 59.35 U/mL produced by the parent strain after submerged fermentation for 10 days, respectively (Figure 4A). In addition, the concentration of citrinin produced by *M. purpureus* LQ-6 was 8.28 mg/L, but citrinin was not detected after the deletion of the *chs*6 gene. Furthermore, the total production of YMPs_-370 nm_, YMPs_-410 nm_, RMPs, and citrinin significantly increased to 45.64 U/mL, 44.30 U/mL, 46.75 U/mL, and 6.65 mg/L after introducing the *chs*6 gene to *M. purpureus* Δ5162. These results illustrate that the *chs*6 gene also plays a vital role in the biosynthesis of secondary metabolites in *Monascus*.

However, among these data, we found that although the total production of MPs was dramatically decreased after the deletion of the *chs*6 gene, the ratios of extracellular YMPs_-370 nm_, YMPs_-410 nm_, and RMPs increased by 22.62%, 10.72%, and 39.23%, respectively (Figure 4B). We speculate that the *Monascus* cell membrane permeability was increased after disrupting the chitin biosynthetic pathway by the deletion of *chs*6, and the overexpression (strong promoter of *gpdA* used in the expression vector) of *chs*6 in *M. purpureus* Δ5162 increased the cell membrane integrity and caused lower production of secondary metabolites than that by the parent strain. Thus, we measured the cytoplasmic leakage of *M. purpureus* Δ5162 and LQ-6. Table 1 shows that the chitin concentration of *M. purpureus* Δ5162 decreased by 27.46% on the 3rd day but increased by 55.56% on the 6th day. Similarly, the concentration of chitin exhibited a 45.5% increase after the deletion of *chs*6 in *B. cinerea*, which resulted from a provoked compensatory mechanism [27]. In addition, our previous study showed that the electrical conductivity of the fermentation broth was significantly increased after the deletion of the *erg*4 gene to increase the cell membrane permeability and enhance MP production and secretion [24]. In this study, the electrical conductivity of *M. purpureus* Δ5162 on the 3rd day and 6th day was dramatically increased by 53.19% and 90.17%, respectively. In addition, it has been reported that AKP is produced in the cytoplasm and leaks into the periplasmic space but is released into the broth from fungal cells with damaged cell walls [31]. In addition, a significantly higher AKP enzyme activity of *M. purpureus* Δ5162 was observed, which was increased by 0.76 and 6.17 times on the 3rd and 6th days, respectively, compared to that of *M. purpureus* LQ-6. These results show that the cell wall and membrane structures were damaged after the deletion of the *chs*6 gene, resulting in higher permeability. However, the gene *chs*6 plays a significant role in the cell growth of *Monascus*, and the severely damaged cellular structures are not conducive to metabolite biosynthesis, which could be the reason that the total MP and citrinin production was significantly reduced.

In addition, the fermentation kinetics (Figure 5) showed that the rates of MPs and citrinin biosynthesis were remarkably reduced after knocking out the *chs*6 gene, compared with that of the wild-type strain. The maximum values of biosynthetic rates of YMPs_-370 nm_, YMPs_-410 nm_, and RMPs_-510 nm_ in submerged fermentation by LQ-6 were 8.68, 9.26, and 8.06 U/mL/d at 8.89, 8.22, and 6.78 days, and their maximum production was 63.87, 74.37, and 59.68 U/mL at 13 days, respectively. However, the deletion of the *chs*6 gene in the LQ-6 strain significantly decreased the maximum values of biosynthetic rates of YMPs_-370 nm_, YMPs_-410 nm_, and RMPs_-510 nm_ to 2.38, 2.68, and 1.58 U/mL/d at 7.44, 8.67, and 8.56 days and their the maximum production to 16.17, 16.67, and 9.22 U/mL at 11 days, respectively. In addition, the citrinin concentration was maintained at approximately zero during submerged fermentation by the *chs*6-deleted strain but was high (10.13 mg/L at 11 days) in the parent strain. However, it has been reported that the yield and productivity of citric acid are improved in submerged fermentation after the silencing of the *chs*C gene in *Aspergillus niger* [32]. Penicillin production is distinctly enhanced by 41% after disturbed *chs4* (class III) gene expression in *P. chrysogenum* [21]. These results illustrate that the regulatory effect of various chitin synthase members on the biosynthesis of target metabolites is diverse in homologous or different microorganisms. Our study showed that *chs*6 has a significantly downregulation effect on the biosynthetic rate of secondary metabolites, especially RMPs.

In fungi, chitin is synthesized using the substrate UDP-N-acetylglucosamine, which can be competitively inhibited by ployoxin B [33,34]. To further confirm the regulatory effect of chitin synthase on secondary metabolism, different concentrations of polyoxin B were added during submerged fermentation. Table 2 shows that when the polyoxin B concentration was lower than 50 mg/L, the MP and citrinin production had minimal changes, but the production of extracellular MPs (exMPs) was enhanced as the polyoxin B concentration increased. Additionally, the production of exYMPs_-370 nm_, exYMPs_-410 nm_, and exRMPs_-510 nm_ was improved by 45.07%, 64.60%, and 56.75%, respectively. However, the MP and citrinin productions were sharply reduced when the polyoxin B concentration was higher than 100 mg/L. The total production of YMPs_-370 nm_, YMPs_-410 nm_, RMPs_-510 nm_, and citrinin decreased by 52.32%, 54.96%, 64.05%, and 95.29% at 200 mg/L and by 68.11%, 73.80%, 81.33%, and 99.51% at 300 mg/L, respectively. In addition, the results showed that the biomass of *M. purpureus* LQ-6 was extremely sensitive to polyoxin B at concentrations >200 mg/L. We speculated that the addition of a low concentration of polyoxin B only increased the cell wall or membrane permeability but did not severely damage the cellular structures. Thus, we compared the gene expression of members of chitin synthase between the exogenous addition of polyoxin B and the deletion of gene *chs*6. In fact, we found that the mRNA levels of the members of chitin synthase were significantly downregulated on day 3 during submerged fermentation after the deletion of *chs6* in *M. purpureus* LQ-6, except for the gene *monascus*_2563 (*chs*A). However, the addition of 50 mg/L of polyoxin B resulted in the downregulated expression of *chs*G, *chs*F, *chs*A, and *chs*2 and slightly upregulated expression of *ch5, chs*E, and *chs*6 on day 3, without significant differences (Table 3). The results show that the effect of *chs*6 deletion on *Monascus* is more significant than the addition of a low concentration of inhibitor polyoxin B in disrupting chitin biosynthesis and demonstrates that *chs6* plays an important role in controlling the biosynthesis of secondary metabolites in *Monascus*.

### 3.4. Comparative Transcriptome Analysis Reveals the Regulation Mechanisms

In this study, a comparative transcriptome analysis of the wild-type strain *M. purpureus* LQ-6 (CK) and mutant strain *M. purpureus* Δ5162 (MT) was performed to reveal the molecular mechanism underlying the regulatory effect of *chs*6 on cell growth and secondary metabolism. The calculation of Pearson correlation coefficients (PCC) was applied to determine the correlations between samples. PCC values of the samples of *M. purpureus* LQ-6 on day 4 and day 7 were 0.971~0.995 and 0.995~0.998, respectively, and those of *M. purpureus* Δ5162 were 0.992~0.998 and 0.987~0.990, respectively. However, PCC values between the two strains at day 4 were 0.757~0.825 and reduced to 0.775~0.813 on day 7 (Figure 6A). These results suggest that the repeated samples have more similar transcriptional programs, but the transcriptional activities between the two strains during submerged fermentation were somewhat distinct. In addition, a total of 8534 genes were acquired, and there were 966 upregulated DEGs and 1698 downregulated DEGs identified in *M. purpureus* Δ5162 on day 4 (CK−1 vs. MT−1) and 1894 upregulated DEGs and 601 downregulated DEGs on day 7 (CK−2 vs. MT−2), compared with the wild-type strain (Figure 6B).

The top 20 terms of GO enrichment analyses showed that the DEGs were mainly involved in oxidoreductase activity (GO:0016491), substrate-specific transmembrane transporter activity (GO:0022891), ion transmembrane transporter activity (GO:0015075), small molecule metabolic processes (GO:0044281), single-organism processes (GO:0044699) in the CK−1 vs. MT−1 group (Figure 7A), and small molecule metabolic processes (GO:0044281), organonitrogen compound metabolic processes (GO:1901564), structural molecule activity (GO:0005198), and hydrolase activity (GO:0016810) in the CK−2 vs. MT−2 group (Figure 7B). In addition, 11 genes (*am*t, *spn*4, *TPS*2, *nim*X, *cap*2, *cpc-*2, *acn*A, *rhp*51, *ftr*A, *ags*1, *vps*24) were upregulated and 15 genes (*byr*2, gene-MPDQ_002275, *sin*1, *myo*A, *ENO*1, *At5g11230*, *KIN7*K, *ags*1, *ste*A, *chs*C, *chs*6, *ssn*3, *PSD*2, *srd5a*3, *cta*1) were downregulated in growth (GO:0040007) in the groups of CK-1 vs. MT-1, and 19 genes (*amt*1, *PMT*4, *SAC*6, *kti*12, *amt*1, *TPS*2, *MYO*2, *csn*D, *bim*G, gene-MPDQ_004737, *mes*A, *cpc-*2, *pda*1, *csn*5, *acn*A, *rhp*51, *ftr*A, *ags*1, *vps*24) were upregulated and 3 genes (*chs*C, *PSD*2, *cta*1) were downregulated in growth (GO:0040007) in the groups of CK−2 vs. MT−2. The top 20 pathways of KEGG pathway enrichment analyses showed that the DEGs were mainly involved in metabolic pathways (ko01100), the biosynthesis of secondary metabolites (ko01110), the biosynthesis of amino acids (ko01230), amino sugar and nucleotide sugar metabolism (ko00520, including genes *chs*G, *chs*D, *chs*B, *chs*C, and *chs*6), and ribosome (ko03010) in the CK−1 vs. MT−1 group (Figure 7C) and in the metabolic pathways (ko01100), the biosynthesis of secondary metabolites (ko01110), carbon metabolism (ko01200), RNA transport (ko03013), aminoacyl-tRNA biosynthesis (ko00970), and proteasome (ko03050) in the CK−2 vs. MT−2 group (Figure 7D).

In this study, we found that cell growth in *Monascus* was significantly regulated by *chs6*. First, we comparatively analyzed the mRNA levels of the various members of chitin synthase between the wild-type and mutant strains to investigate the effect of the *chs6* gene on the expression of other genes. The results showed that the expression of *chs*C, *chs*D, and *chsG* was significantly downregulated, which was also the case for the key enzymes involved in chitin biosynthesis, including glucosamine-phosphate N-acetyltransferase (EC 2.3.1.4) and UDP-N-acetylglucosamine diphosphorylase (EC 2.7.7.23), while that of the others was slightly decreased in *M. purpureus* Δ5162 on day 4 compared with that in the parent strain. However, the DEGs on day 7 were irregular; only the *chs*C gene was significantly downregulated by 4-fold, and the expression levels of others were weakly increased or decreased in the mutant strain, compared with those of the wild-type strain, but the gene encoding glucosamine-phosphate N-acetyltransferase was significantly downregulated. Further, DEGs analysis showed that the global regulator *Lae*A and developmental regulatory protein *Wet*A were significantly downregulated by more than 3- and 34-fold after knocking out the *chs6* gene in *M. purpureus* LQ-6 on day 4, but not on day 7, during submerged fermentation. Zhang et al. reported that the pitting degree, number of particles, and folding degree of mycelia could be significantly increased by the overexpression of *Lae*A in *M. purpureus* [18]. In addition, the *Lae*A-deleted *Monascus* strain exhibits abundant aerial hyphae and no typical cleistothecia [19]. During the growth period (4 days), the transcription factors involved in the development (bZIP-like protein, *C2H2*, *crz*A, *nos*A, *FCR*3, *lep*B, *MBZ*1, Zn(II)2Cys6, *Flb*B, *YGR266*W, *BRG*1, late sexual development protein, serotype 2.1 precursor, *pil*2, *cet*A, and *ncs*1), the MAPK signaling pathway (including 16 DEGs), and the genes involved in cell wall structures (*ENO*1, GPI-anchored cell wall protein, *cet*A, *fae*B-2, *mok*13, and cell wall biogenesis protein) were significantly downregulated. Signaling pathways, including mitogen-activated protein kinase (MAPK) signaling pathways in fungi, are highly conserved and closely related to the cell-wall integrity of filamentous fungi [35]. GPI-anchored cell wall proteins are required for normal cell growth, and they perform various functions and interact with the environment [36]. These downregulated pathways explain the above experimental phenomena of inhibited conidia germination and differentiation, altered mycelial morphology, reduced stress tolerance, and decreased cell wall thickness due to the deletion of the *chs*6 gene in *Monascus*.

The biosynthesis of MPs and citrinin was significantly inhibited after the disruption of the chitin biosynthesis pathway by the deletion of the *chs*6 gene in *Monascus* during submerged fermentation (Figure 8). The genes *MPsGe*A, *MPsGe*B, *MPsGe*L, *MPsGe*I, *MPsGe*H, *MPsGe*G, *MPsGe*F, *MPsGe*E, *MPsGe*D, *MPsGe*C, *MPsGe*P, MPDQ_006010, *MPsGe*O, *MPsGe*N, *MPsGe*M, *MPsGe*K, and *MPsGe*J in the MP biosynthetic gene cluster were significantly downregulated by more than 4.13, 2.85, 2.05, 2.53, 4.39, 3.91, 5.41, 6.33, 3.10, 4.61, 4.19, 8.34, 6.78, 6.77, 4.04, 6.18, and 7.76 times, respectively, at the fermentation prophase (day 4) (Figure 8A). Although they still maintained the downregulated mRNA levels at the logarithmic stage of fermentation (day 7), only *MPsGe*L (ankyrin repeat protein) and gene-MPDQ_006010 (decarboxylase) were significantly downregulated by 2.94 and 2.25 times, respectively. The expression levels of other genes were not significantly different. However, the expressions of the DEGs in the citrinin biosynthetic gene cluster were sharply decreased by 8–25 times (except *cnt*F and *cnt*R) at the fermentation prophase and more severely downregulated (more than 16–83 times) at the logarithmic stage of fermentation (Figure 8B). This may explain the difference in MP and citrinin production during submerged fermentation by the mutant strain and illustrates that *chs*6 could regulate the biosynthesis of secondary metabolites in *Monascus*, especially citrinin.

From Figure 9, except for the downregulated expression levels of genes located in the genetic clusters (MPs and citrinin biosynthesis), the related pathways, such as fatty acid synthesis (FAS), fatty acid degradation (FAD), tricarboxylic acid cycle (TCA), and polyketone synthesis (PKS), also showed some different changes during the SBF process. Most obviously, the expression level of the gene encoding pyruvate decarboxylase (EC:4.1.1.1) and the FAD pathway were upregulated on day 7 (which were downregulated at day 4), driving the biosynthesis of acetyl-CoA and stimulating the FAS and PKS pathway to contribute to the total anaplerotic production of MPs and citrinin. Furthermore, the increased cell wall permeability from the disruption of chitin biosynthesis enhanced the MPs secretion, causing the overproduction of extracellular MPs and complementing the expression of genes located in the cluster. However, the low production of total MPs may be due to the significantly reduced concentration of precursors and biomass. Overall, at the fermentation prophase, the cell growth phenotypes of *M. purpureus* LQ-6, including spore germination, hypha differentiation, and mycelium morphology, were synergistically controlled by MAPK signaling pathway, regulatory factors *Lae*A and *Wet*A, and GPI anchored cell-wall protein. At the logarithmic stage of fermentation, the secondary metabolism was regulated by the pathways of FAS, FAD, PKS, and the corresponding genetic clusters. In addition, the impaired cell growth would provoke amino acid synthesis, nucleotide sugar metabolism, carbon metabolism, and ribosome, and autophagy would be inhibited because of the upregulated proteasome after deletion of *chs*6 gene, which promises the normal cell growth of the mutant strain at the logarithmic stage of fermentation.

## 4. Conclusions

Secondary metabolites of *Monascus* have been studied for centuries in Asia. Researchers have found that the biosynthesis of metabolites is closely related to the morphology of filamentous fungi, and the disruption of the chitin synthesis pathway can significantly change mycelial morphology during fermentation. In the present study, a *chs6*-deleted mutant strain was generated using PEG-mediated protoplast transformation. Image analysis revealed that the *chs6* gene regulates development, cell wall structures, and mycelial morphology and is required for the normal growth of *Monascus*. In addition, the MP biosynthesis pathway was strongly inhibited, and almost no citrinin was produced after the deletion of *chs*6. Furthermore, the molecular mechanism underlying the regulatory effect of *chs*6 on cell growth and secondary metabolism in *M. purpureus* was elucidated by comparative transcriptome analysis.

## Figures and Tables

**Figure 1 jof-08-00910-f001:**
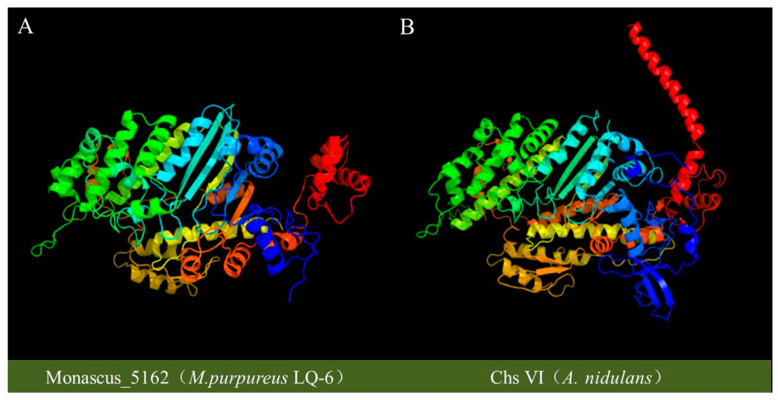
The predicted tertiary structure of protein *monascus*_5162 from *M. purpureus* LQ-6 (**A**) and *chs*6 from *A. nidulans* (**B**).

**Figure 2 jof-08-00910-f002:**
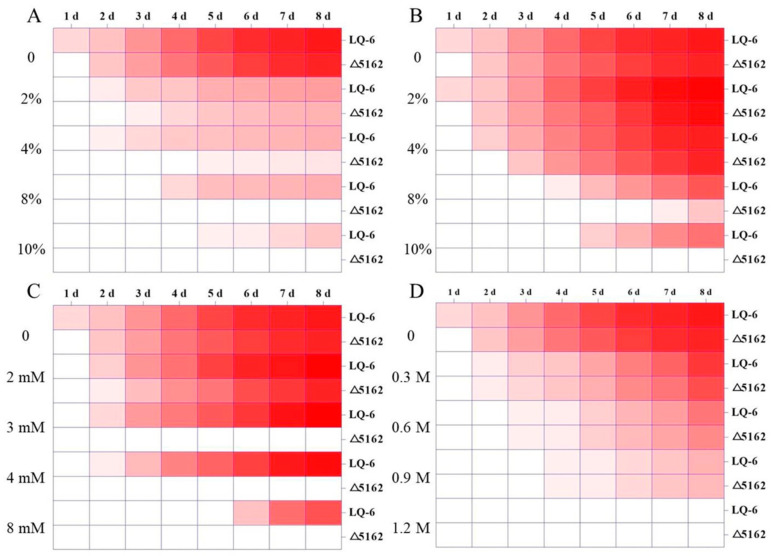
The sensitivities of parent strain *M. purpureus* LQ-6 and *chs*6-deleted strain *M. purpureus* Δ5162 cultured on PDA medium at 30 °C in the dark to lactic acid (**A**), ethanol (**B**), H_2_O_2_ (**C**), and NaCl (**D**). Red color represents normal cell growth, white color represents the cell growth is completely inhibited.

**Figure 3 jof-08-00910-f003:**
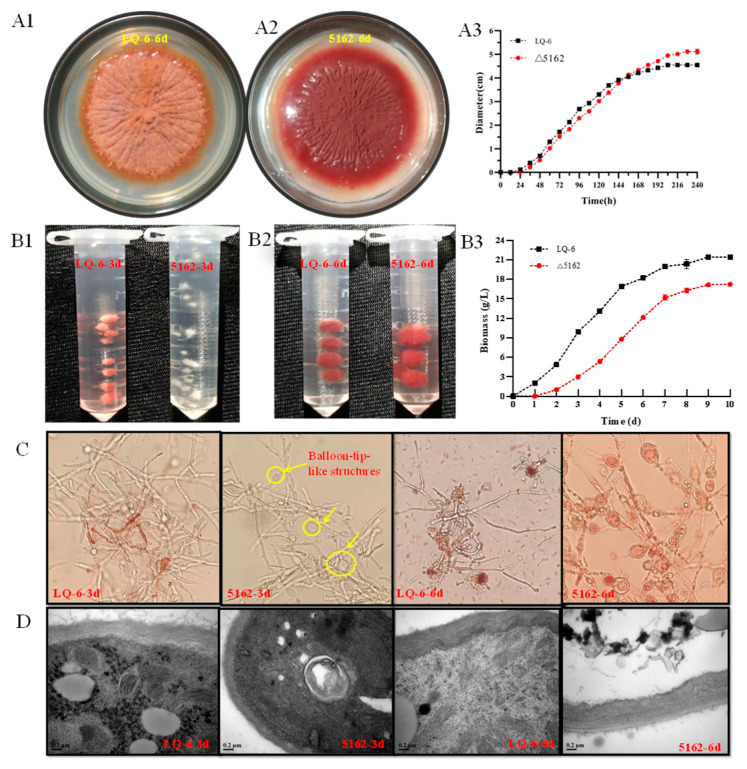
Image analysis of the cell growth phenotypes of *M. purpureus* LQ-6 and *M. purpureus* Δ5162. The colonial morphology of *M. purpureus* LQ-6 (**A1**) and *M. purpureus* Δ5162 (**A2**) on PDA medium, and the colony diameter (**A3**); the mycelial pellets morphologies at 3 day (**B1**) and 6 days (**B2**) during submerged fermentation, and the biomass (**B3**); the microscopic analysis of mycelium in submerged fermentation (**C**); transmission electron micrographs of the cell wall structure (**D**).

**Figure 4 jof-08-00910-f004:**
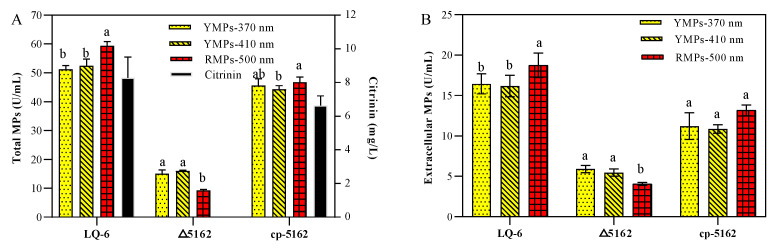
The total *Monascus* pigments and citrinin production (**A**), and extracellular *Monascus* pigments production (**B**) by the parent strain *M. purpureus* LQ-6 and recombination strains in submerged fermentation for 10 days at 30 °C and agitated at 150 rpm in the dark. Different letters (a and b) up the columns in the group of same strain are considered statistically different (*p* < 0.05), but the columns marked a or b to ab are considered insignificant difference.

**Figure 5 jof-08-00910-f005:**
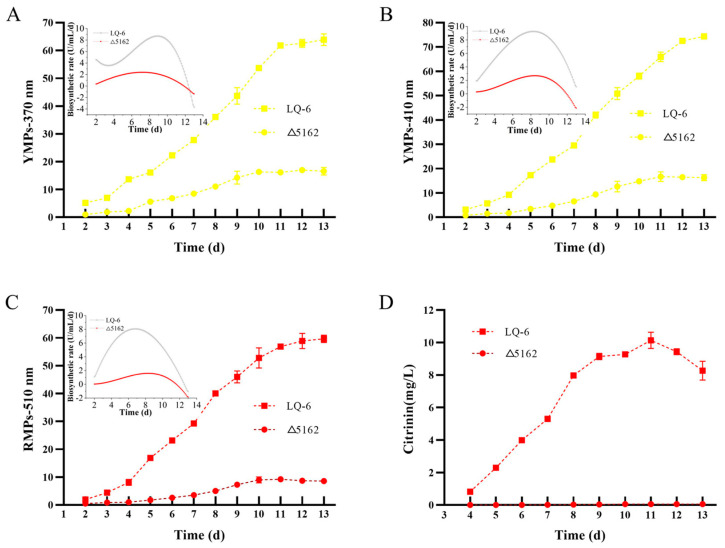
Submerged batch-fermentation kinetics analysis of *M. purpureus* LQ-6 and *M. purpureus* Δ5162. The biosynthesis of YMPs-370 nm (**A**), biosynthesis of YMPs-410 nm (**B**), biosynthesis of RMPs-510 nm (**C**), and biosynthesis of citrinin (**D**).

**Figure 6 jof-08-00910-f006:**
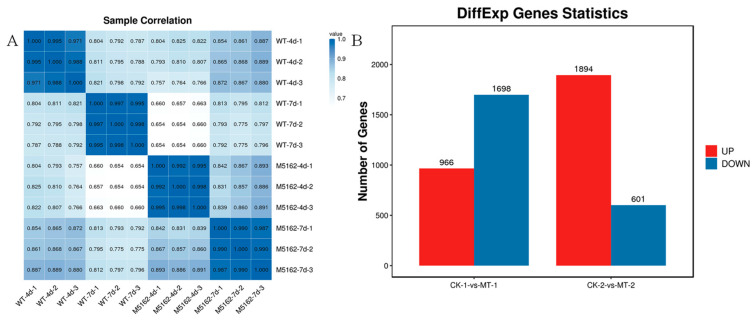
Pearson correlation coefficient analysis of RNA-seq data from parent strain and mutant strain (**A**), differentially expressed genes between these two strains at various stages of submerged fermentation (**B**).

**Figure 7 jof-08-00910-f007:**
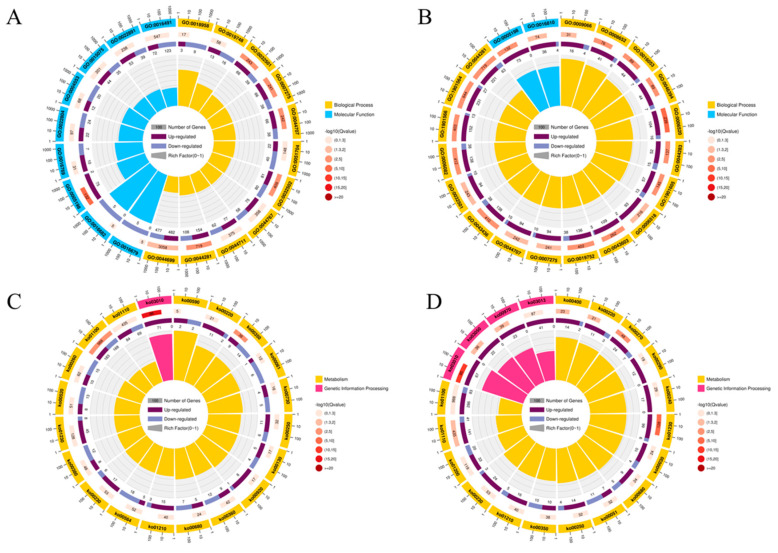
The Gene Ontology (GO) enrichment analyses in the CK−1 vs. MT−1 group (**A**) and CK−2 vs. MT−2 group (**B**), and Kyoto Encyclopedia of Genes and Genomes (KEGG) pathway enrichment analyses in the CK−1 vs. MT−1 group (**C**) and CK−2 vs. MT−2 group (**D**) of differentially expressed genes.

**Figure 8 jof-08-00910-f008:**
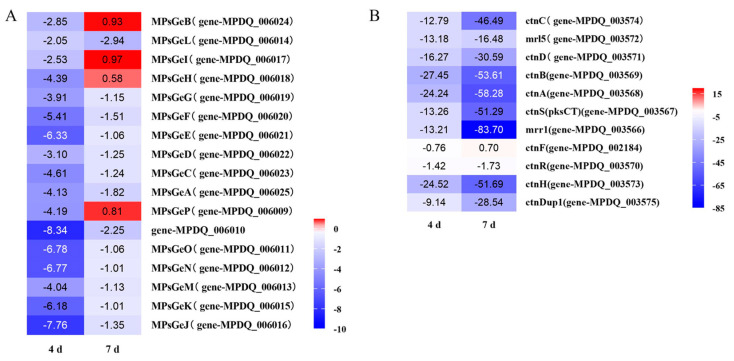
The changes in mRNA levels of *Monascus* pigments’ biosynthetic gene cluster (**A**), and citrinin biosynthetic gene cluster (**B**) after knocking out the *chs*6 gene in the parent strain *M. purpureus* LQ-6 during the submerged fermentation at the fermentation prophase (day 4) and the logarithmic stage of fermentation (day 7).

**Figure 9 jof-08-00910-f009:**
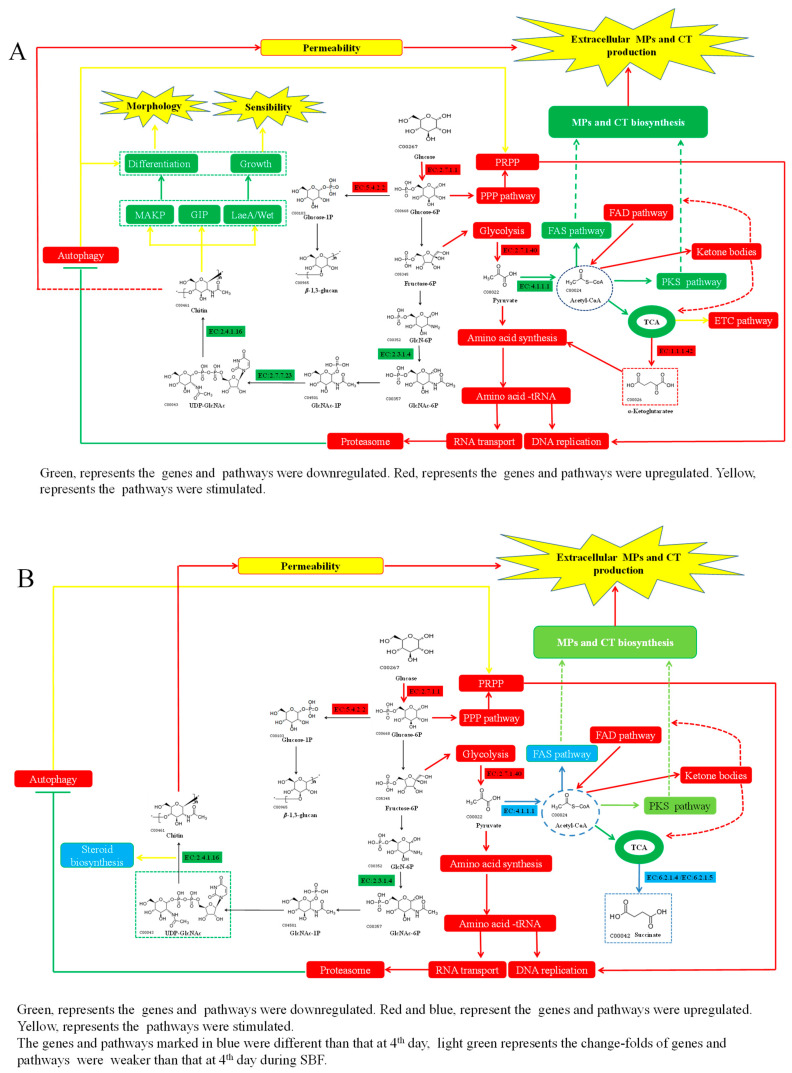
The regulated pathways after knockout of the *chs*6 gene in parent strain *M. purpureus* LQ-6 during submerged fermentation at the fermentation prophase (**A**) and logarithmic stage of fermentation (**B**).

**Table 1 jof-08-00910-t001:** Determination of *M. purpureus* LQ-6 and *M. purpureus* Δ5162 strains for chitin content and cell wall and membrane integrity.

Strain	Chitin Concentration (mg/g)		Electrical Conductivity (μs/cm)		AKP Activity (U/g pro)	
	3 d	6 d	3 d	6 d	3 d	6 d
*M. purpureus* LQ-6	45.30 ± 2.12	40.03 ± 1.75	457.6 ± 56.0	618.4 ± 37.7	0.198 ± 0.012	0.247 ± 0.028
*M. purpureus* Δ5162	32.86 ± 3.22	62.27 ± 6.93	701.0 ± 11.3	1176.0 ± 21.1	0.348 ± 0.046	1.772 ± 0.136

**Table 2 jof-08-00910-t002:** The effect of ployoxin B on biomass, *Monascus* pigments and citrinin production of *M. purpureus* LQ-6 in submerged fermentation for 10 days at 30 °C in the dark. Different letters in the same column indicate a significant difference (*p* < 0.05).

Polyoxin B (mg/L)	YMPs- 370 nm (U/mL)	YMPs- 410 nm (U/mL)	RMPs- 510 nm (U/mL)	exYMPs- 370 nm (U/mL)	exYMPs- 410 nm (U/mL)	exRMPs- 510 nm (U/mL)	Citrinin (mg/L)	Biomass (g/L)
0	51.15 ± 0.21 a	52.75 ± 3.46 b	59.30 ± 1.41 a	16.44 ± 1.06 cd	16.16 ± 0.21 b	18.73 ± 0.42 c	8.28 ± 0.14 b	21.73 ± 0.82 a
20	55.70 ± 3.67 a	54.30 ± 3.67 b	52.15 ± 2.02 b	17.80 ± 1.27 bc	18.24 ± 0.85 b	22.52 ± 2.47 b	8.08 ± 0.22 b	21.37 ± 0.52 ab
50	55.45 ± 2.47 a	63.60 ± 1.69 a	61.90 ± 3.25 a	23.85 ± 0.64 a	26.60 ± 1.57 a	29.36 ± 0.42 a	10.28 ± 0.25 a	19.81 ± 0.88 c
100	39.53 ± 2.65 b	42.28 ± 2.72 c	38.07 ± 2.17 c	19.45 ± 0.85 b	18.52 ± 2.23 b	18.89 ± 1.41 c	3.57 ± 0.42 c	20.27 ± 0.53 bc
200	24.39 ± 3.70 c	23.76 ± 3.68 d	21.32 ± 3.15 d	14.15 ± 0.46 d	12.08 ± 0.41 c	12.15 ± 0.46 d	0.39 ± 0.11 d	18.05 ± 0.35 d
300	16.31 ± 0.15d	13.82 ± 0.42 e	11.07 ± 1.20 e	9.62 ± 0.47 e	6.70 ± 0.39 d	5.46 ± 0.39 e	0.04 ± 0.03 d	17.31 ± 0.43 d

**Table 3 jof-08-00910-t003:** Comparison of the gene expression levels of different members of chitin synthesis at 3 days during submerged fermentation by addition of 50 mg/L of ployoxin B and deletion of *chs*6 gene in *M. purpureus* LQ-6.

Gene ID	Description	Mean Log2 Ratio	
		50 mg/L Ployoxin B	△5162
*Monascus*_2400	*Chs* III (*chs*G)	−0.49	−2.35
*Monascus*_2508	*Chs* IV(*chs*F)	−0.31	−2.04
*Monascus*_2563	*Chs* II (*chsA*)	−0.25	0.22
*Monascus*_2765	*Chs* V (*chs*5)	0.17	−1.19
*Monascus*_4382	*Chs* II *chs*2	−0.17	−1.39
*Monascus*_5161	*Chs* VII (*chsE*)	0.45	−2.35
*Monascus*_5162	*Chs* VI (*chs*6)	0.41	-

## Data Availability

The data presented in this study are available upon request from the corresponding author. All raw RNA-seq data were deposited in the NCBI database using the BioProject accession number: PRJNA785330. The gene *monascus_5162* is located in the genome sequence of *M. purpureus* LQ-6 with the accession number PRJNA503091.

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
