# Peer review of "Disruption of the Chitin Biosynthetic Pathway Results in Significant Changes in the Cell Growth Phenotypes and Biosynthesis of Secondary Metabolites of Monascus purpureus"

_jof, 2022, doi:10.3390/jof8090910_

Round 1

Reviewer 1 Report

Please a deep discussion of the gene monascus-5162 from Monascus purpureus LQ-6, identified as chitin synthase gene VI (chs6), was knocked out to disrupt the chitin biosynthetic pathway and regulate the biosynthesis of Monascus pigments (MPs) and citrinin mechanism.
The results showed that the aerial hyphae on solid medium were short and sparse after deletion of chs6 in M. purpureus LQ-6, significantly reducing the germination percentage of active spores to approximately 22%, but the colony diameter was almost unaffected. Additionally, the deletion of chs6 changed the mycelial morphology of M. purpureus LQ-6 during submerged fermentation and increased its sensitivity to environmental factors. Although the presented data seems interesting, the no usage the statistical analyses rises a red flag for me and precludes me from interpreting the presented data in a professional manner. I suggest that the authors redo their statistical analyses before this manuscript can be published. My major concerns are the lack of statistical information and proper testing.

Author Response

Please a deep discussion of the gene monascus-5162 from Monascus purpureus LQ-6, identified as chitin synthase gene VI (chs6), was knocked out to disrupt the chitin biosynthetic pathway and regulate the biosynthesis of Monascus pigments (MPs) and citrinin mechanism.
The results showed that the aerial hyphae on solid medium were short and sparse after deletion of chs6 in M. purpureus LQ-6, significantly reducing the germination percentage of active spores to approximately 22%, but the colony diameter was almost unaffected. Additionally, the deletion of chs6 changed the mycelial morphology of M. purpureus LQ-6 during submerged fermentation and increased its sensitivity to environmental factors. Although the presented data seems interesting, the no usage the statistical analyses rises a red flag for me and precludes me from interpreting the presented data in a professional manner. I suggest that the authors redo their statistical analyses before this manuscript can be published. My major concerns are the lack of statistical information and proper testing.

Response: Thank you very much for your question. Firstly, we have added the statistical analyses in the revised manuscript, all the experimental data obtained at least in triplicate, furtherly, we marked the significant difference in figures and table 2. Secondly, there are many interesting phenomenon after deletion of gene chs6 in M. purpureus LQ-6, and the results data showed that gene chs6 play a vital role in cell growth and secondary metabolism, including MPs and ctrinin. In addition, we also found that chitin synthase pathway, global regulator LaeA and developmental regulatory protein WetA, transcription factors, MAPK signaling pathway and MPs and CT biosynthetic gene clusters and related genes both have significantly changes. Besides, we have made a discussion and summarize in Fig. 9 to indicate the regulation mechanism by chs6 gene in M. purpureus LQ-6 from the comparative transcriptome analysis. 

Reviewer 2 Report

According to the manuscript, Disruption of the chitin biosynthetic pathway results in significant changes in the cell growth phenotypes and biosynthesis of secondary metabolites of Monascus purpureus. This is an intriguing investigation of the function of the chs6 gene in Monascus purpureus. The investigation's goal is self-evident. However, it has a critical flaw that must be fixed and the paper's structure must be restructured. Please find the reviewer’s comments below.

1.    In section 2.4, the authors should draw the construction of gene deletion and complementation, and plasmid.

2.    Line 112, change the liquid PDA medium to PDB medium.

3.    Add a section on statistical analysis to the method and describe the method the authors used as well as the replication of the experiment.

4.    As the authors describe in lines 217-219, the authors used the protein chitin synthase from Aspergillus terreus NIH2624 please check Fig.1 which presented the A. nidulans data and in the figure legends also (line 226).

5.    Please indicate the difference in the protein structure in Fig. 1.

6.    Please review the wording on lines 239-241. Do you wish to discuss the LQ-6 or mutant germination time? (Because the authors wrote as after knocking out chs6).

7.    The authors should explain the meaning of the color in Fig.2. in the figure legends.

8.    To easily understand the sensitivity of fungus to each chemical, the sensitivities studied in Fig. 2 should present the same concentration units (%).

9.    As the authors mention in lines 282-288, where is the result data of the germination rates of the conidia? Please add these data in supplementary files.

10. As the authors mention in Fig. 3C, the mutant strain shows some tip balloon-like structures in the hyphae that appeared on day 3. Please indicate it on the figure.

11. I confused Fig. 4 result, what is the difference between Total MPs and Extracellular MPs in Fig 4 A and B? Did you measure it from the fermentation broth?

12. At the Y axis of Fig 4B, please change the excellular to extracellular.

13. As the authors describe on lines 331-333, What are the ratios of extracellular YMPs-370 nm, YMPs-410 nm, and RMPs (Fig 4B)? How is it increase?

14. Please correct Fig.9 to the horizontal position for better detail.

15.  In the result and discussion, most of them are just experimental results. There was little discussion of the results. The authors should separate between results and discussion and increase the discussion of experimental results.

Author Response

According to the manuscript, Disruption of the chitin biosynthetic pathway results in significant changes in the cell growth phenotypes and biosynthesis of secondary metabolites of Monascus purpureus. This is an intriguing investigation of the function of the chs6 gene in Monascus purpureus. The investigation's goal is self-evident. However, it has a critical flaw that must be fixed and the paper's structure must be restructured. Please find the reviewer’s comments below.

1.    In section 2.4, the authors should draw the construction of gene deletion and complementation, and plasmid.
Response 1: Thank you very much for your question. We have added them in Fig. S1, and supplemented the according sentence “These construction processes are showed in Fig. S1” in section 2.4.

  1. Line 112, change the liquid PDA medium to PDB medium.
    Response 2: Thank you very much for your question. We have revised it.

  2. Add a section on statistical analysis to the method and describe the method the authors used as well as the replication of the experiment.
    Response 3: Thank you very much for your question, and sorry for our carelessness. We have added the section of “2.14. Statistical analysis” in the revised manuscript.

  3. As the authors describe in lines 217-219, the authors used the protein chitin synthase from Aspergillus terreus NIH2624 please check Fig.1 which presented the A. nidulans data and in the figure legends also (line 226).
    Response 4: Thank you very much for your question, and sorry for our carelessness. The reference gene or protein is located in A. nidulans not A. terreus NIH2624, and we have revised it to “Besides, the protein chitin synthase from Aspergillus nidulans FGSC A89, CsmB (BAE78841), is annotated as chs6 (https://www.ncbi.nlm.nih.gov/protein/86355213).”

  4. Please indicate the difference in the protein structure in Fig. 1.
    Response 5: Thank you very much for your question. They are almost same, and we have described the difference in the manuscript (the result showed that the spatial structure of proteins of them were almost same, except for longer C-terminal tail in CsmB and 2 hits in monascus_5162). In fact, the length of monascus_5162 and CsmB(chs6) is 1755 aa and 1739 aa, respectively, and CLUSTAL W (1.83) multiple sequence alignment showed that Identities is 65%, Positives is 76%, Gaps is 3%.

  5. Please review the wording on lines 239-241. Do you wish to discuss the LQ-6 or mutant germination time? (Because the authors wrote as after knocking out chs6).

Response 6: Thank you very much for your question. Yes, we showed the difference between parent strain LQ-6 and the gene chs-deleted strain, and we found that the germination time of M. purpureus Δ5162 was prolonged to 4th day, but the parent strain just need 2 days.

7.    The authors should explain the meaning of the color in Fig.2. in the figure legends.
Response 7: Thank you very much for your suggestion, we have added it in the legend of Fig.2 .

  1. To easily understand the sensitivity of fungus to each chemical, the sensitivities studied in Fig. 2 should present the same concentration units (%).

Response 8: Thank you very much for your suggestion. Many researches major in cell tolerance to factors, including ethanol, H2O2, NaCl and acid. Generally, the liquid chemical, especially ethanol and lactic acid, were often presented by using percentage; NaCl, as a solid chemical compound, but is generally partial to “Mol” unit. Besides, in this manuscript, although the presented units are different, the differences between parent strain and chs6-deleted strain are dramatically obvious.      

9.    As the authors mention in lines 282-288, where is the result data of the germination rates of the conidia? Please add these data in supplementary files.
Response 9: Thank you very much for your question. We have added it in Table S3.

  1. As the authors mention in Fig. 3C, the mutant strain shows some tip balloon-like structures in the hyphae that appeared on day 3. Please indicate it on the figure.
    Response 10: Thank you very much for your question. We have indicated them in Fig.3C.

  2. I confused Fig. 4 result, what is the difference between Total MPs and Extracellular MPs in Fig 4 A and B? Did you measure it from the fermentation broth?
    Response 11: Thank you very much for your question. The fermentation supernatant was directly measured for analysis of extracellular MPs (Fig. 4B), the total MP concentrations (Fig.4A) include intracellular MPs concentrations and extracellular MPs concentrations, which have been described in our previous study, the reference 24. In addition, we have revised the section of “2.11. Determination of metabolites” to “The processing and measurement of citrinin and MPs content, residual glucose concentration, biomass, and electrical conductivity were conducted according to our previous report [24].”.

  3. At the Y axis of Fig 4B, please change the excellular to extracellular.
    Response 12: Thank you very much for your question, and sorry for our carelessness, we have revised it.

  4. As the authors describe on lines 331-333, What are the ratios of extracellular YMPs-370 nm, YMPs-410 nm, and RMPs (Fig 4B)? How is it increase?

Response 13: Thank you very much for your question. From Fig. 4B, to LQ-6, the ratio of extracellular YMPs-370 nm was 32.13% (=16.44/51.165); to the mutant strain, the ratio of extracellular YMPs-370 nm was 39.40% (=5.91/15); then, the ratio of extracellular YMPs-370 nm increased by 22.62% [=(39.40%-32.13%)/32.13%],and same as others.

14. Please correct Fig.9 to the horizontal position for better detail.

Response 14: Thank you very much for your suggestion, but they are horizontal (Fig. 9A and 9B).

15.  In the result and discussion, most of them are just experimental results. There was little discussion of the results. The authors should separate between results and discussion and increase the discussion of experimental results.

Response 15: Thank you very much for your question. The combination of result and discussion is allowable in this journal. Besides, we also showed many discussions of the results in this manuscript, especially in “3.4. Comparative transcriptome analysis reveals the regulation mechanisms”. 
